# Exploring Loss Functions for Time-based Training Strategy in Spiking Neural Networks

**Yaoyu Zhu**
School of Computer Science
Peking University
Beijing, China 100871
yaoyu.zhu@pku.edu.cn

**Wei Fang**
School of Computer Science
Peking University
Beijing, China 100871
fwei@pku.edu.cn

**Tiejun Huang**
School of Computer Science
Peking University
Beijing, China 100871
tjhuang@pku.edu.cn

**Xiaodong Xie**
School of Computer Science
Peking University
Beijing, China 100871
donxie@pku.edu.cn

**Zhaofei Yu** [*]
Institute for Artificial Intelligence
Peking University
Beijing, China 100871
yuzf12@pku.edu.cn

## Abstract

Spiking Neural Networks (SNNs) are considered promising brain-inspired energy-efficient models due to their event-driven computing paradigm. The spatiotemporal spike patterns used to convey information in SNNs consist of both rate coding and temporal coding, where the temporal coding is crucial to biological-plausible learning rules such as spike-timing-dependent-plasticity. The time-based training strategy is proposed to better utilize the temporal information in SNNs and learn in an asynchronous fashion. However, some recent works train SNNs by the time-based scheme with rate-coding-dominated loss functions. In this paper, we first map rate-based loss functions to time-based counterparts and explain why they are also applicable to the time-based training scheme. After that, we infer that loss functions providing adequate positive overall gradients help training by theoretical analysis. Based on this, we propose the enhanced counting loss to replace the commonly used mean square counting loss. In addition, we transfer the training of scale factor in weight standardization into thresholds. Experiments show that our approach outperforms previous time-based training methods in most datasets. Our work provides insights for training SNNs with time-based schemes and offers a fresh perspective on the correlation between rate coding and temporal coding. Our code is available at https://github.com/zhuyaoyu/SNN-temporal-training-losses.

## 1 Introduction

Artificial Neural Networks (ANNs) have achieved remarkable progress in various fields, such as computer vision [23], natural language processing [58], and reinforcement learning [51]. Compared with the conventional non-spiking ANNs, Spiking Neural Networks (SNNs) exhibit a greater level of biological plausibility [65] and achieve significantly improved energy efficiency on neuromorphic chips [9, 17, 37, 44, 49]. The benefit of SNNs on computation cost and power consumption originates from their event-driven nature to a large extent [47].

ANNs use real values to encode the state of a neuron. This encoding scheme is similar to the firing rate of a biological neuron. Inspired by this encoding scheme, researchers have applied ANN-SNN

---

[*]Corresponding author

37th Conference on Neural Information Processing Systems (NeurIPS 2023).

conversion to obtain the parameters of SNNs [7, 22, 52]. Unlike firing rate coding, temporal coding is a unique feature of SNNs (compared with ANNs), which utilizes the temporal information of spikes in SNNs and encodes information by the absolute and relative firing times between spikes. The most commonly used temporal coding in SNNs is time-to-first-spike (TTFS) coding [3, 39, 67], which encodes the information by the firing time of the first spike emitted by a neuron. Although the encoding schemes in SNNs are richer than ANNs, directly training SNNs is harder than ANNs due to the binary nature of spikes and the non-differentiable membrane potential at spike time. Therefore, SNNs has not reached the performance of their ANN counterparts in classification tasks [10, 63].

There are two major categories of supervised learning techniques to train SNNs: The first category is activation-based learning, which treats SNNs as binary-output recurrent neural networks (RNNs) and uses backpropagation-through-time (BPTT) to train SNNs [2, 16, 19, 59]. The main difference between training SNNs and RNNs with these methods is that they apply continuous surrogate derivatives to substitute the discontinuous spike firing process [40]. Although competitive performances are achieved on the CIFAR-10/100, and even the ImageNet dataset [11, 14], these methods do not follow the asynchronous nature of SNNs. Firing-rate-based coding is intensively used in these approaches. In the output stage, the commonly used loss functions include spike-train-based ones [50, 64], spike-count-based ones [59, 60], and those defined on discrete time steps [11, 15]. All these loss functions behave in a rate coding scheme and have no direct relation to spike firing time (which indicates temporal coding). The second category is spike-time-based learning, which considers spikes' timing as an information carrier for parameter update in SNNs [3, 8, 61]. They are more biologically plausible since they keep the event-driven nature of biological learning rules [69]. Originated in SpikeProp [3], researchers try to solve the discontinuity at spike firing time and gradient vanishing/explosion in backpropagation. In the output stage, most of them use time-based loss functions [38, 66], which is consistent with the temporal coding. However, loss functions not belonging to the temporal coding family can also successfully train SNNs with time-based gradients [67, 69]. This phenomenon has not yet been clearly explained.

In this work, we investigate the relationship between rate coding and temporal coding by analyzing the loss functions that correspond to both coding schemes applied in time-based training algorithms. Besides, we study how to improve the performance of time-based training methods with appropriate loss functions (as well as other mechanisms). Our main contributions can be summarized as follows:

1. We prove that rate-based loss functions, extensively used in activation-based SNN training, suit the more biologically plausible time-based training scheme. This result implies that there are implicit relations between rate coding and temporal coding in SNNs.
2. We find that loss functions providing adequate positive overall gradients are more suitable for time-based training schemes in SNN training. Based on this, we propose the enhanced counting loss to improve the previously used mean square counting loss.
3. We transfer the training of scale factor used by weight standardization into thresholds. Experiments have shown that this action improves network performance.
4. We test our method on MNIST, Fashion-MNIST, NMIST, CIFAR10, CIFAR100, DVS-Gesture, and CIFAR10-DVS datasets. The proposed method achieves SOTA performance around time-based SNN training methods on most datasets.

## 2   Related Work

**Neuron coding scheme.** Spike timing information provides SNNs with richer coding schemes than non-spiking ANNs, which can be grouped into three main types: rate, temporal, and burst. Rate coding encodes information by firing rates of neurons [57], which is widely used in SNN training [7, 40]. Temporal coding is shown to convey richer and more precise information than rate coding [35, 53, 54], which can be classified into time-to-first-spike (TTFS) [55], rank-order [56], and phase coding [26]. Among them, the most commonly used scheme is the TTFS coding [3, 39, 43, 67], which can be applied to both the network input and neuron output. When applied to network input, it converts static pixel intensity into spike firing time (higher intensity corresponds to earlier input spike time) [66, 68]. When encoding the neuron outputs, it takes the timing of the first spike a neuron fires as the feature of a neuron [3, 68]. Besides, several recent works explore burst coding, a group of short inter-spike interval spikes, to improve the efficiency of information transmission and enhance the performance of SNNs [42]. Some works also extend the temporal-based rank-order coding into ANNs [25], bringing forward the time of decisions of ANNs without trading off accuracy.

**Training methods of SNNs.** SNN learning methods mainly include ANN-SNN conversion and direct training. ANN-SNN conversion assists SNN training with ANNs, which trains an ANN and replaces its activation functions by spiking neurons with transformed weights to get an SNN [7]. The converted SNN can achieve comparable performance as the source ANN when simulated long enough [5, 12, 21, 33, 48], whereas degrading seriously when the number of time steps is small [6]. Direct training mainly falls into two categories. The first category is activation-based training, which regards SNNs as RNNs and trains them with the Backpropagation Through Time (BPTT) method [11, 13–16, 20, 34, 36, 59, 60, 64]. Surrogate gradients [40] are used in this scheme to handle the discontinuity when neurons emit spikes. SNNs trained in this fashion can be scaled to large datasets like ImageNet [14] with low latency. They can deal with temporal tasks such as classifying neuromorphic datasets [11] while relying on rate coding: The loss function and gradient propagation scheme modify synaptic weights by modifying the firing rate of neurons. Specifically, the loss functions they use includes spike-train-based losses [50, 64], spike-count-based losses [59, 60], and those defined on discrete time steps [11, 15], among which losses defined on discrete time steps empirically perform better. The second category is time-based training, which takes the spike firing times as the intermediate variable to update synaptic weights [3]. Most algorithms in this track approximate the derivative of output spike timing to membrane potential $\frac{dt_{out}}{du}$ as the negative inverse of membrane potential's time derivative $\frac{-1}{du/dt}$ [3, 4, 38, 66, 67, 69]. Some works use other approximations [27], while others directly derive the relation between output spike firing times and input spike firing times [8, 39]. Although this training scheme has not yet trained SNNs with comparable performance as BPTT-based approaches, they satisfy the asynchronous nature of SNNs [69] and better utilize temporal coding in SNNs. In addition to the spike timing of neurons used in the backward calculation, the loss functions adopted in this scheme are typically time-based losses [3, 38, 39, 66]. However, other loss functions can also successfully train SNNs with time-based gradients [67, 69]. This leaves a question: Why can rate-coding-based losses successfully be applied in the time-based training scheme? We will analyze this question in the following content.

## 3 Preliminaries

**Spike response model.** In this work, we use the spike response model for the neurons in the Spiking Neural Network. The state of neurons can be described by the following equations [3, 18, 67]:

$$u_i^{(n)}(t) = \int_{t_{i,\text{last}}^{(n)}}^{t} \left( \sum_j w_{ij}^{(n)} \cdot s_j^{(n-1)}(\tau) \right) \cdot \epsilon(t - \tau) d\tau, \tag{1}$$

$$s_i^{(n)}(t) = \delta(u_i^{(n)}(t) - \theta), \tag{2}$$

where $u_i^{(n)}(t)$ denotes the membrane potential of neuron $i$ in layer $n$ at time $t$, $w_{ij}^{(n)}$ denotes the weight between neuron $j$ in layer $n-1$ and neuron $i$ in layer $n$. $t_{i,\text{last}}^{(n)}$ is the time of last spike of neuron $i$ in layer $n$, and $s_i^{(n)}(t)$ represents the spike emitted from neuron $i$ at time $t$. The function $\delta(\cdot)$ is the Dirac Delta function and $\theta$ is the firing threshold. The spike response kernel is $\epsilon(t) = \frac{\tau_m}{\tau_m - \tau_s}(e^{-\frac{t}{\tau_m}} - e^{-\frac{t}{\tau_s}})$, where $\tau_m$ and $\tau_s$ are membrane and synapse time constants, respectively. Following [67, 69], we eliminate the influence of input spikes prior to the last output spike on membrane potentials.

**Backpropagation based on spike timing.** The backpropagation formulas in the time-based training scheme are given by [69]:

$$\frac{\partial L}{\partial w_{ij}^{(n)}} = \sum_{i, t_k(s_i^{(n)})} \frac{\partial L}{\partial t_k(s_i^{(n)})} \frac{\partial t_k(s_i^{(n)})}{\partial u_i^{(n)}(t_k)} \frac{\partial u_i^{(n)}(t_k)}{\partial w_{ij}^{(n)}}, \tag{3}$$

$$\frac{\partial L}{\partial t_m(s_j^{(n-1)})} = \sum_{i, t_k(s_i^{(n)})} \frac{\partial L}{\partial t_k(s_i^{(n)})} \frac{\partial t_k(s_i^{(n)})}{\partial u_i^{(n)}(t_k)} \frac{\partial u_i^{(n)}(t_k)}{\partial t_m(s_j^{(n-1)})}, \tag{4}$$

where $L$ is the loss function, $t_k(s_i^{(n)})$ and $t_m(s_j^{(n-1)})$ are timings of spikes $s_i^{(n)}$ (by neuron $i$ in layer $n$) and $s_j^{(n-1)}$ (by neuron $j$ in layer $n-1$), $u_i^{(n)}(t_k)$ is the membrane potential of neuron $i$ in layer $n$ at time $t_k$, and $w_{ij}^{(n)}$ is the synaptic weight between neuron $i$ in layer $n$ and neuron $j$ in layer $n-1$.

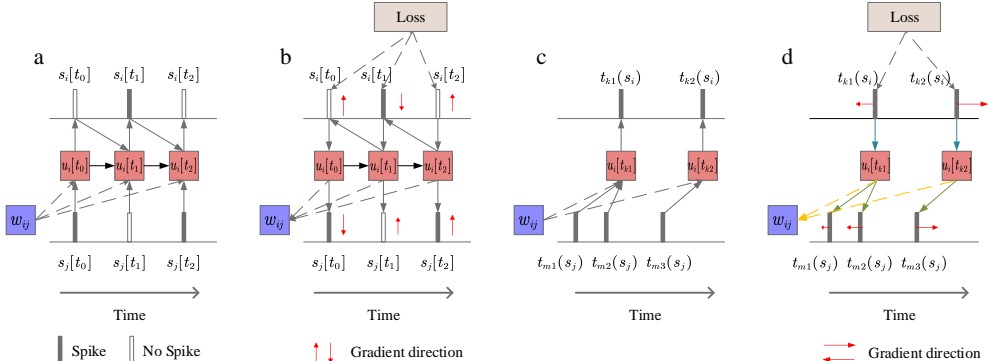

Figure 1: Forward and backward processes in activation-based training schemes and time-based training schemes. (a) Forward and (b) backward propagation in activation-based schemes: In the backward stage, the loss function updates parameters to adjust the firing rate. (c) Forward and (d) backward propagation in time-based schemes: In both forward and backward stages, information is delivered by spikes, which accords with the asynchronous nature of SNNs. The parameters are updated to adjust the firing times in the backward stage.

Figure 1(c)(d) shows the gradient propagation path in Eq. (3) and (4). It starts from the loss function and propagates to spike firing times $t_k(s_i)$ of neurons, neuron membrane potentials at spike times $u_i(t_k)$, and synaptic weights $w_{ij}$ in each layer. There are three major derivatives: output spike timing to membrane potential $\frac{\partial t_k(s_i^n)}{\partial u_i^n(t_k)}$, membrane potential to input spike timing $\frac{\partial u_i^n(t_k)}{\partial t_k(s_j^{n-1})}$, and membrane potential to weight $\frac{\partial u_i^n(t_k)}{\partial w_{ij}}$. The latter two can be directly derived since the original function (Eq. (1)) is continuous. The first derivative needs to take an approximation due to discontinuity at spike time:

$$\frac{\partial t_k(s_i^n)}{\partial u_i^n(t_k)} = \frac{-1}{du_i^n(t_k)/dt},\tag{5}$$

which is first proposed by Spikeprop [3] and followed by many other works [38, 67, 69]. It has been proved that the sum of time-based gradients does not change between layers during backpropagation based on the above gradient calculation approach [69]:

$$\sum_i \sum_{t_k} \frac{\partial L}{\partial t_k(s_i^{(n)})} = \sum_j \sum_{t_m} \frac{\partial L}{\partial t_m(s_j^{(n-1)})}.\tag{6}$$

Therefore, for time-based gradients, the derivative of spike firing time with respect to the loss function is equipped with more useful properties than the loss function itself. In other words, it is natural to view the loss function from the aspect of the gradients on spike firing times.

## 4 Methods

Here we first establish a framework to relate the gradients of the spike train to spike firing time for the output layer in Sect. 4.1 and 4.2. Starting with this framework, we will show how to view rate-based losses (defined on spike trains) as time-based losses (defined on spike firing times) with two examples, the rate-based counting loss used by [69] and spike train loss used by [67]. Sect. 4.3 analyzes the influence of overall gradients on spike timing on the SNN performance. Based on our analysis, we infer that loss functions providing adequate positive overall gradients are more suitable for time-based training schemes. Formulated by this inference, we propose the enhanced counting loss in Sect. 4.4. In Sect. 4.5, we propose to transfer the training of scale factor used by weight standardization into thresholds since threshold values do not appear in backpropagation formulas. At last, we will provide an overall algorithm description in Sect. 4.6.

### 4.1 Relate Gradients of Spike Train to Spike Firing Time

Unlike time-based losses defined on spike firing times, a rate-based loss $L$ is defined on the output spike train $s$ instead of the firing times of the output spike train $t(s)$. As a result, the spike train itself

should be an intermediate variable to pass the gradients, and a derivative from $\boldsymbol{s}$ to $\boldsymbol{t}(\boldsymbol{s})$ should be defined in the backpropagation stage:

$$\frac{\partial L}{\partial \boldsymbol{t}(\boldsymbol{s})} = \frac{\partial L}{\partial \boldsymbol{s}} \cdot \frac{\partial \boldsymbol{s}}{\partial \boldsymbol{t}(\boldsymbol{s})}. \tag{7}$$

Note that $\frac{\partial L}{\partial \boldsymbol{s}}$ is defined by the loss function itself, but $\frac{\partial \boldsymbol{s}}{\partial \boldsymbol{t}(\boldsymbol{s})}$ is somewhat custom. Hence we add limitations on the derivative $\frac{\partial \boldsymbol{s}}{\partial \boldsymbol{t}(\boldsymbol{s})}$: First, gradients should not propagate from one spike to the timing of another spike, which means $\frac{\partial \int s_i(t)}{\partial t'(s_{i'})} \neq 0$ only if $i = i'$ and $t = t'$. Here $\int s_i(t)$ means converting an infinite Dirac delta spike $s_i(t)$ to a finite value 1. Hence the derivative of the timing of one spike is

$$\frac{\partial L}{\partial t_k(s_i)} = \frac{\partial L}{\partial \int s_i(t_k)} \cdot \frac{\partial \int s_i(t_k)}{\partial t_k(s_i)}. \tag{8}$$

Second, noticing there are still two degrees of freedom ($\frac{\partial L}{\partial \int s_i(t_k)}$ and $\frac{\partial \int s_i(t_k)}{\partial t_k(s_i)}$) when calculating the gradient, we can control one of them to set $\frac{\partial \int s_i(t_k)}{\partial t_k(s_i)} = 1$.

However, for ease of understanding, we will allow $\frac{\partial \int s_i(t_k)}{\partial t_k(s_i)}$ to take value $-1$ in Sect. 4.2 and 4.4.

## 4.2 Why Can Rate-coded Losses be Applied in The Time-based Training Scheme?

Actually, rate-based loss functions have their time-based counterparts. For rate-based loss $L$, there is a time-based loss $\mathcal{L}$ that has the same derivation, that is, $\frac{\partial \mathcal{L}}{\partial t_k(s_i)} = \frac{\partial L}{\partial t_k(s_i)}$. We first consider the case when loss functions do not involve spike timing in their expressions and have the following theorem:

**Theorem 4.1.** *For a rate-based loss function $L(\boldsymbol{s})$ which does not involve spike timing, with a spike-to-timing derivative $\frac{\partial \boldsymbol{s}}{\partial \boldsymbol{t}(\boldsymbol{s})} = diag(\frac{\partial \int s_i(t_k)}{\partial t_k(s_i)}) = I$ where $(s_i, t_k)$ takes all neurons and spikes, it has an equivalent form $\mathcal{L}(\boldsymbol{t}(\boldsymbol{s})) = \sum_i \sum_{t_k(s_i)} \frac{\partial L}{\partial s_i(t_k)} \cdot t_k(s_i)$ defined on the timing of spikes.*

The detailed proof is provided in the Appendix. With Theorem 4.1, we can analyze why rate-based losses can be applied in the time-based training scheme.

**Counting loss.** The counting loss is commonly used for activation-based gradients [59, 60] satisfies the condition in Theorem 4.1. It has the form:

$$L(\boldsymbol{s}, \textbf{target}) = \lambda \sum_{i=1}^{C} \left( \int_0^T s_i(t)dt - \text{target}_i \right)^2. \tag{9}$$

Here $\lambda$ is a scaling constant, and $C$ is the number of classes (which equals the number of output neurons). Besides, **target** is the target spike count sequence, and $\text{target}_i$ is the target spike number fired by neuron $i$. Typically, $\text{target}_i$ is bigger when $i$ corresponds to the label. According to Theorem 4.1, the equivalent time-based loss function of the counting loss (when all $\frac{\partial \int s_i(t_k)}{\partial t_k(s_i)} = 1$) is

$$\mathcal{L}(\boldsymbol{t}(\boldsymbol{s})) = \sum_i \sum_{t_k(s_i)} \frac{\partial L}{\partial \int s_i(t_k)} \cdot t_k(s_i) = \sum_i \sum_{t_k(s_i)} 2\lambda \left( \int_0^T s_i(t)dt - \text{target}_i \right) \cdot t_k(s_i). \tag{10}$$

The detailed deduction is provided in the Appendix. The equivalent temporal-coded loss $\mathcal{L}$ is linear for each spike firing time $t_k(s_i)$. It encourages a spike to fire earlier when the number of fired spikes of the corresponding neuron exceeds the target ($\frac{\partial \mathcal{L}}{\partial t_k(s_i)} > 0$ when $\int_0^T s_i(t)dt - \text{target}_i > 0$) and fire later when the spike count is below the target. This is actually contrary to our expectations. The simplest way to fix it is modifying $\frac{\partial \int s_i(t_k)}{\partial t_k(s_i)} = 1$ to $\frac{\partial \int s_i(t_k)}{\partial t_k(s_i)} = -1$, which is adopted by [69]. As a result, our analysis framework not only explains why counting losses works for time-based training but also shows why $\frac{\partial \int s_i(t_k)}{\partial t_k(s_i)} = -1$ is necessary.

**Spike train difference loss.** The difference between spike trains [46] is also widely used in activation-based SNNs [29, 50, 64]. It can be described by the following formula:

$$L(\boldsymbol{s}, \boldsymbol{s}^{\text{target}}) = \lambda \sum_{i=1}^{C} \int_0^T \left( \kappa * (s_i - s_i^{\text{target}})(t) \right)^2 dt. \tag{11}$$

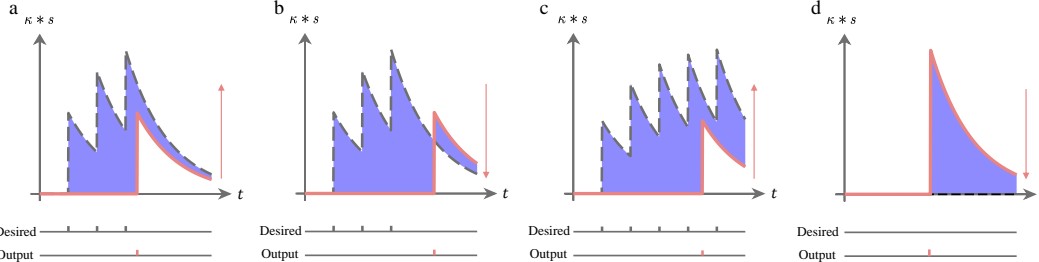

Figure 2: The spike train difference loss. It aims to minimize the difference (purple area) between the (filtered) output spike train (red curve) and the target spike train (dashed curve). The kernel here is $\kappa(t) = \alpha \cdot e^{-\alpha t}$. (a)(b)(c)(d) are four different situations, where the output is pushed upward in (a)(c) and downward in (b)(d) by the loss. Note that the desired output spike train is empty in (d).

Here, $s^{\text{target}}$ is the target spike train for output neurons, and $s_i^{\text{target}}$ is the target spike train of neuron $i$. Typically the target spike train of the neuron corresponding to the label contains more spikes than other neurons. $\kappa$ is a 1-d convolution kernel to smooth the spike trains. Theorem 4.1 can't directly apply due to spike timings being involved in the loss function, making it difficult to find its corresponding temporal-coded function. Instead, we apply qualitative analysis to it.

The spike train difference loss measures the height difference between two filtered spike trains, as shown in Figure 2. Sometimes we can find inconsistencies for it in the time-based training scheme: In both Figure 2 (a)(b), the spike should be moved earlier on the time axis. However, spikes will be moved in different directions (minimizing the purple area by moving up or down since it acts on spike scale $s_i(t_k)$) in two cases under this loss. This spike direction inconsistency will impede training. On the other hand, if the target spike train is dense enough and covers all time ranges (Figure 2 (c)), the spike will always be pushed upward when the output spike train is sparse. Meanwhile, it is clear that output spikes of a neuron will be pushed down when the target spike train is empty (Figure 2 (d)). In both cases (c)(d), there are no inconsistencies. As a result, we can also correspond turning spike scale up with moving spike time earlier and let $\frac{\partial \int s_i(t_k)}{\partial t_k(s_i)} = -1$ in this case. Our analysis explains why spike train difference loss works for time-based training and why $\frac{\partial \int s_i(t_k)}{\partial t_k(s_i)} = -1$ is adopted in [67].

### 4.3 Balance Positive and Negative Time-based Gradient

According to Eq. (6), the positive part and the negative part of time-based gradients should be balanced to control the sum of gradients since they keep unchanged among layers.

**Deficiencies of counting loss.** The derivative $\frac{\partial \int s_i(t_k)}{\partial t_k(s_i)} = -1$ will cause the sum of gradients on spike timing to comprise too much positive portion near convergence, especially when the target number of spikes for label neurons is large. To see this, according to Eq. (8), supposing a label output neuron emits $m$ spikes, then the total gradients on these spikes are:

$$\sum_{s_i(t_k)|i=\text{label}} \frac{\partial L}{\partial t_k(s_i)} = m \frac{\partial L}{\partial t_{k_0}(s_i)} = 2\lambda m(\text{target}_{\text{label}} - m). \tag{12}$$

Here 'label' is the index of the labeled neuron, and $t_{k_0}(s_i)$ is a representative spike timing since all $\frac{\partial L}{\partial t_k(s_i)}$ are equal. The detailed proof is provided in the Appendix. This gradient sum on label output neurons is a quadratic function of the number of output spikes $m$, which provides too much positive gradient on spike timing when $m$ is large.

**Positive sum of gradients on spike timing.** When the network is nearly converged, output neurons corresponding to the labels often emit much more spikes than other neurons, which implies that the slope of membrane potential $\frac{du}{dt}$ for these neurons at spike time is usually more significant than others. As a result, the $\frac{\partial t}{\partial u}$ part on the gradient propagation path (refer to Figure 1) for label neurons is smaller than others in scale (but their signs are both negative). If the sum of gradients is exactly zero, the weights in the last layer will receive more positive gradients than negative ones, leading to decreasing weights in the last layer. This effect will inhibit output neurons from emitting spikes.

## 4.4 Enhanced Counting Loss

Based on the analysis in Section 4.3, we infer that loss functions providing adequate positive overall gradients are more suitable for time-based training schemes. Building upon this conclusion, we introduce a novel loss function to optimize network training. Since the counting loss (used in [69]) provides too much positive gradient on spike timing when the label output neuron emits many spikes, it is more adequate to let $\sum_{s_i(t_k)} \frac{\partial L}{\partial t_k(s_i)}$ be a linear function $2\lambda(\text{target}_i - m)$ instead of a quadratic function $2\lambda m(\text{target}_i - m)$ in Eq. (12). To do this, suppose a neuron fires $m$ spikes, we can substitute $\frac{\partial L(m)}{\partial m} = 2(m - \text{target}_i)$ with $\frac{\partial L(m)}{\partial m} = \frac{2(m-\text{target}_i)}{m}$. The form of this modified loss function is

$$L(\boldsymbol{s}, \textbf{target}) = 2\lambda \sum_{i=1}^{C} f\left(\int_0^T s_i(t)dt, \text{target}_i\right). \tag{13}$$

$$\text{where } f(x, \text{target}) = \begin{cases} x - \text{target}\ln(x), & x > 0 \\ 0, & x \leq 0 \end{cases} \tag{14}$$

with derivative $\frac{\partial \int s_i(t_k)}{\partial t_k(s_i)} = -1$. We name the loss as **enhanced counting loss**. The detailed equation-deducing procedure is provided in the Appendix.

## 4.5 Moving Scaling on Weights to Thresholds

In previous works, weight standardization [45] is used [69] to stabilize training since common normalization schemes [24] are not applicable in asynchronous networks. The weights of a layer are decided by three sets of parameters $\tilde{W}, \gamma, \beta$, which correspond to the original weight, scale, and shift:

$$\hat{W}^{(k)} = \frac{\tilde{W}^{(k)} - E[\tilde{W}^{(k)}]}{\sqrt{\text{Var}[\tilde{W}^{(k)}] + \varepsilon}}, \quad W^{(k)} = \gamma^{(k)} \cdot \hat{W}^{(k)} + \beta^{(k)}. \tag{15}$$

Here $x^{(k)}$ means the $k$-th channel of $x$, $\hat{W}^{(k)}$ is the normalized weights for $\tilde{W}^{(k)}$, and the final weight $W^{(k)}$ goes through a linear transform with the scale $\gamma^{(k)}$ and shift $\beta^{(k)}$. In time-based training (Figure 1(c)(d)), weights participate in backpropagation by passing gradients from output spike timing to membrane potential and then to input spike timing of a neuron. According to Eqs. (2) (5), we have:

$$\frac{\partial t_k(s_i^{(l)})}{\partial u_i^{(l)}(t_k)} = \frac{-1}{du_i^{(l)}(t_k)/dt} = \left(\sum_{j, t_{k,\text{last}}(s_i^{(l)}) < t_m(s_j^{(l-1)}) \leq t_k(s_i^{(l)})} \frac{\partial u_i^{(l)}(t_k)}{\partial t_m(s_j^{(l-1)})}\right)^{-1}, \tag{16}$$

$$\frac{\partial u_i^{(l)}(t_k)}{\partial t_m(s_j^{(l-1)})} = w_{ij}^{(l)} \cdot \frac{\partial \epsilon(t_k - t_m)}{\partial t_m}. \tag{17}$$

Therefore the change in the weight scale will influence the gradient backpropagation by changing $w_{ij}^{(l)}$ in Eq.17. However, threshold values do not appear in the above backpropagation paths. Thus, learning thresholds instead of weights might stabilize the training process. In forward formulas Eqs. (1) (2), scaling up weights $w_{ij}^{(n)}$ and thresholds $\theta$ by the same factor will not change the output spike train. The weight scale is determined by $\gamma$ and $\beta$, so we can transform to keep $\gamma$ constant and learn $\theta$:

$$\theta^{(k)} \leftarrow \frac{\gamma_0}{\gamma^{(k)}} \cdot \theta^{(k)}, \beta^{(k)} \leftarrow \frac{\gamma_0}{\gamma^{(k)}} \cdot \beta^{(k)}, \gamma^{(k)} \leftarrow \gamma_0, \tag{18}$$

where $\gamma_0$ is the value we want to fix $\gamma$ to. When doing parameter updates, denote $d\gamma^{(k)} = \frac{\partial L}{\partial \gamma^{(k)}}$ and $d\beta^{(k)} = \frac{\partial L}{\partial \beta^{(k)}}$, the parameter update rule after the above transformation can be described by:

$$d\theta_{\text{new}}^{(k)} = d(\frac{\gamma_0}{\gamma^{(k)}} \theta^{(k)}) = \theta^{(k)} \frac{-\gamma_0}{(\gamma^{(k)})^2} d\gamma^{(k)} = -\frac{\theta^{(k)}}{\gamma^{(k)}} d\gamma^{(k)}, \tag{19}$$

$$d\beta_{\text{new}}^{(k)} = d(\frac{\gamma_0}{\gamma^{(k)}} \beta^{(k)}) = \frac{\gamma_0}{\gamma^{(k)}} d\beta^{(k)} + \beta^{(k)} \frac{-\gamma_0}{(\gamma^{(k)})^2} d\gamma^{(k)} = d\beta^{(k)} - \frac{\beta^{(k)}}{\gamma^{(k)}} d\gamma^{(k)}. \tag{20}$$

It should be noticed that the value of $\gamma^{(k)}$ will be unchanged (always equal to $\gamma_0$) during learning after this transformation. During learning, parameters to be learned are $\theta$ and $\beta$ where $\theta^{(k)} \leftarrow \theta^{(k)} + \eta d\theta_{\text{new}}^{(k)}$ and $\beta^{(k)} \leftarrow \beta^{(k)} + \eta d\beta_{\text{new}}^{(k)}$.

## 4.6 Overall Algorithm

---

**Algorithm 1** Discrete Version of Our Algorithm

---

**Input:** $s^{(0)}$: Input spike train, $\boldsymbol{W}^{(n)}(n = 1, ..., N)$: Weight for all layers (where $N$ is the number of layers), Total simulation time $T$
**Output:** $\nabla \boldsymbol{W}^{(n)}(n = 1, ..., N)$
// Forward propagation:
Initialize $\boldsymbol{Neuron}^{(n)}(n = 1, ..., N)$: Neurons with states (membrane potential parameters) for all layers
**for** $n = 0$ to $N - 1$ **do**
    **for** $t = 0$ to $T - 1$ **do**
        Update state of $\boldsymbol{Neuron}^{(n+1)}$ according to Eq. 1
        **if** Membrane potential $\boldsymbol{u}^{(n+1)}[t] \geq \theta$ **then**                                ▷ Fire spikes
          $\boldsymbol{s}^{(n+1)}[t] = 1$
          Reset state of $\boldsymbol{Neuron}^{(n+1)}$
        **end if**
    **end for**
**end for**
// Backward propagation:
Initialize $\nabla \boldsymbol{W}^{(n)} \leftarrow \boldsymbol{0}$, $\frac{\partial L}{\partial \boldsymbol{t}(s)^n} \leftarrow 0$ for $n = 1, 2, ..., N$
Calculate $\frac{\partial L}{\partial \boldsymbol{t}(s)^N}$ according to Eq. 13 14
**for** $n = N$ to $1$ **do**
    **for** $t = T - 1$ to $0$ **do**
        **if** $\boldsymbol{s}^{(n)}(t) = 1$ **then**                                    ▷ Spikes fired
          Set $\frac{\partial L}{\partial \boldsymbol{u}^{(n)}}[t]$ to $\frac{\partial L}{\partial \boldsymbol{t}(s^{(n)})} \frac{\partial \boldsymbol{t}(s^{(n)})}{\partial \boldsymbol{u}^{(n)}(t)}$     ▷ $\frac{\partial L}{\partial \boldsymbol{t}(s^{(n)})} \frac{\partial \boldsymbol{t}(s^{(n)})}{\partial \boldsymbol{u}^{(n)}(t)}$ is part of Eq. 3 4
          Reset $\Delta \boldsymbol{t} \leftarrow 0$
        **else**
          Set $\Delta \boldsymbol{t} \leftarrow \Delta \boldsymbol{t} + 1$
        **end if**
        // $\frac{\partial \boldsymbol{u}^{(n)}}{\partial \boldsymbol{W}^{(n)}}$ and $\frac{\partial \boldsymbol{u}^{(n)}}{\partial \boldsymbol{t}(s^{(n-1)})}$ both involve $\epsilon(\Delta \boldsymbol{t})$ where $\epsilon(\cdot)$ is the spike response kernel
        Set $\nabla \boldsymbol{W}^{(n)} \leftarrow \nabla \boldsymbol{W}^{(n)} + \frac{\partial L}{\partial \boldsymbol{u}^{(n)}} \cdot \frac{\partial \boldsymbol{u}^{(n)}}{\partial \boldsymbol{W}^{(n)}}$                 ▷ Eq. 3
        Set $\frac{\partial L}{\partial \boldsymbol{t}(s^{(n-1)})} \leftarrow \frac{\partial L}{\partial \boldsymbol{u}^{(n)}} \cdot \frac{\partial \boldsymbol{u}^{(n)}}{\partial \boldsymbol{t}(s^{(n-1)})}$             ▷ Eq.4
    **end for**
    Update $\boldsymbol{\theta}, \boldsymbol{\beta}$ according to Eq. 19 20 as well as $\tilde{\boldsymbol{W}}$ using $\nabla \boldsymbol{W}^{(n)}$
**end for**

---

Algorithm 1 describes our overall algorithm. Our algorithm works in discrete time-steps here since it suits better for current deep learning frameworks such as PyTorch. The main training and inference workflow follows [69]. The enhanced counting loss is in the beginning of the backward propagation and the transfer of training from scaling factor to threshold is embodied in the second last line.

## 5 Experiments

We conduct experiments on MNIST [31], Fashion-MNIST [62], NMNIST [41], CIFAR10/100 [30], DVS-Gesture [1], and CIFAR10-DVS [32] datasets to evaluate the performance of our method. The methods we compare are Spiking Neural Networks trained with time-based gradients carried by spikes, including TC [39], ASF [8], S4NN [27], BS4NN [28], STiDi-BP [38], STDBP [66], TSSL-BP[67], and EDBP [69]. All experiments are run on single Nvidia RTX 3080Ti/3090 GPUs. The implementation details are provided in the Appendix.

### 5.1 Compare with the State-of-the-Art

We first compare our method with current state-of-the-art time-based SNN training approaches. Table. 1 reports the results on MNIST, Fashion-MNST, N-MNIST, CIFAR10, CIFAR100, DVS-

Table 1: Performance comparison on MNIST and CIFAR-10/100 datasets

| Dataset | Model | Gradient Type | Architecture | Accuracy |
|---------|-------|---------------|--------------|----------|
| MNIST | TC [39] | Temporal | 784-800 | 97.5% |
| | ASF [8] | Temporal | 784-340 | 97.9% |
| | S4NN [27] | Temporal | 784-600 | 97.4% |
| | BS4NN [28] | Temporal | 784-600 | 97.0% |
| | STiDi-BP [38] | Temporal | 784-500 | 97.4% |
| | STDBP [66] | Temporal | 16C5-P2-32C5-P2-800-128[a] | 99.4% |
| | TSSL-BP [67] | Temporal | 15C5-P2-40C5-P2-300 | **99.53%** |
| | EDBP [69] | Temporal | 15C5-P2-40C5-P2-300 | 99.47% |
| | ANN | - | 15C5-P2-40C5-P2-300 | 99.50% |
| | **Ours** | Temporal | 15C5-P2-40C5-P2-300 | **99.50%** |
| Fashion-MNIST | S4NN [27] | Temporal | 784-1000 | 88.0% |
| | BS4NN [28] | Temporal | 784-1000 | 87.3% |
| | STDBP [66] | Temporal | 16C5-P2-32C5-P2-800-128 | 90.1% |
| | TSSL-BP [67] | Temporal | 32C5-P2-64C5-P2-1024 | 92.83% |
| | EDBP [69] | Temporal | 32C5-P2-64C5-P2-1024 | 93.28% |
| | ANN | - | 32C5-P2-64C5-P2-1024 | 93.94% |
| | **Ours** | Temporal | 32C5-P2-64C5-P2-1024 | **94.03%** |
| N-MNIST | TSSL-BP [67] | Temporal | 12C5-P2-64C5-P2 | **99.40%** |
| | EDBP [69] | Temporal | 12C5-P2-64C5-P2 | 99.39% |
| | **Ours** | Temporal | 12C5-P2-64C5-P2 | **99.39%** |
| CIFAR10 | TSSL-BP [67] | Temporal | CIFARNet[b] | 91.41% |
| | EDBP [69] | Temporal | VGG11[b] | 92.10% |
| | ANN | - | VGG11 | 94.51% |
| | **Ours** | Temporal | VGG11 | **93.54%** |
| CIFAR100 | EDBP [69] | Temporal | VGG11 | 63.97% |
| | ANN | - | VGG11 | 74.23% |
| | **Ours** | Temporal | VGG11 | **70.50%** |
| DVS-Gesture | **Ours** | Temporal | VGG11 | **97.22%** |
| CIFAR10-DVS | **Ours** | Temporal | VGG11 | **76.30%** |

[a] 15C5: convolution layer with 15 channels of $5 \times 5$ filters; P2: pooling layer with $2 \times 2$ filters
[b] The detailed network architectures are provided in the Appendix (section Experimental Settings)

Gesture, and CIFAR10-DVS datasets. It should be noticed that we do not list BPTT-based methods here since they make use of information when neurons do not fire spikes in the gradient propagation, which certainly benefits the training process and the final performance. On the easy MNIST and N-MNIST datasets, the performance of our algorithm is comparable to those of TSSL-BP [67] and EDBP [69]. The improvement is not noticeable as the performance is saturated ($> 99\%$). On other datasets, our algorithm evidently outperforms other time-based SNN training algorithms. For the Fashion-MNIST dataset, our algorithm outperforms the previous time-based SOTA algorithm [69] by 0.75%. For the CIFAR10 dataset, we achieve 93.54% top-1 accuracy with a VGG11 network, outperforming EDBP [69] by 1.44% with the same network architecture. Besides, we have achieved 70.50% accuracy on the CIFAR100 dataset, outperforming the previous SOTA SNN trained with a time-based scheme by more than 6% with the same architecture. In addition, we are the first time to train SNNs with time-based schemes on both the DVS-Gesture and CIFAR10-DVS datasets, achieving 97.22% accuracy on the DVS-Gesture dataset and 76.30% accuracy on the CIFAR10-DVS dataset. All these results demonstrate that the proposed method outperforms the state-of-the-art accuracy on nearly all datasets.

## 5.2 Ablation Study

In this section, we study the effects of our proposed loss function and normalization scheme on the network performance. We select EDBP [69] as the baseline in our ablation experiments, which uses counting loss and weight normalization. The performance of SNNs with different loss functions and normalization schemes is illustrated in Table. 2 and Figure 3(a). It can be seen that the proposed enhanced counting loss outperforms the original counting loss on CIFAR100 to a large extent. We achieve 68.95% top-1 accuracy, which is higher than the baseline by 4.97%. At the same time, our proposed threshold normalization scheme is also effective in increasing the network performance.

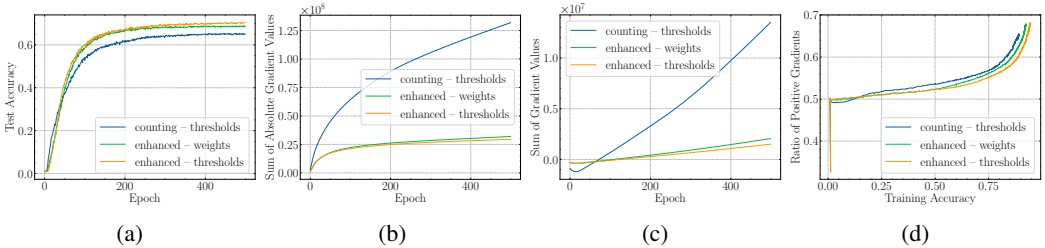

|        (a)        |        (b)        |        (c)        |        (d)        |

Figure 3: Ablation results. (a) Test accuracy along with training epoch index. (b) Accumulated sum of absolute gradient values along with training epoch index. (c) Accumulated sum of gradient values along with training epoch index. (d) The ratio of positive gradients along with training accuracy. In the figures, the **blue line** indicates the counting loss and threshold normalization scheme; the **green line** indicates the enhanced counting loss and weight normalization scheme; the **orange line** indicates the enhanced counting loss and threshold normalization scheme.

To further illustrate how our proposed methods help the network training, we analyze the gradient scale as well as the ratio of positive gradients. Figure 3(b) plots the gradient scale described by the $L1$-norm of gradients, which is the sum of the absolute gradient value. Also, we provide the sum of (not absolute) gradient values in Figure 3(c). Both values are plotted against the

Table 2: Ablation study on the CIFAR-100 dataset

| Loss Function | Normalization | Accuracy |
|---|---|---|
| Count (MSE) | Weight | 63.98% |
| Count (MSE) | Threshold | 66.35% |
| Enhanced | Weight | 68.95% |
| **Enhanced** | **Threshold** | **70.50**% |

number of training epochs, and they are both accumulated through epochs. From the figures, we can find that the enhanced counting loss greatly reduces the overall scale of gradients. Besides, the proposed threshold normalization scheme also slightly reduces the overall gradient scale.

We also show the ratio of positive gradients among all gradients in Figure. 3(d). It should be noticed that we do not plot it against the epoch index, instead, we plot it against the network training accuracy. The underlying reason is that this ratio is largely influenced by the ratio between the number of spikes that label output neurons emitted and other output neurons emitted. The spike emission status comparison between label neurons and other neurons in the output layer can be better described by the network training accuracy compared with the epoch index, especially when the convergence rates in different settings are different. Besides, we do not accumulate gradient data in this figure since we want to get the information in each epoch. From this figure, we can see that in most cases, gradients trained with the enhanced counting loss and the threshold normalization scheme are more balanced (closer to 0.5 in the figure). There is a rising tendency for the proportion of positive gradients with the convergence of the network, which is influenced by more spikes emitted by label output neurons. In the very beginning, the negative part of gradients is much larger than the positive part. Oppositely, the overall gradients mainly consist of the positive part near convergence.

## 6 Conclusion

In this paper, we demonstrate the applicability of rate-coding-based loss functions to time-based training. We analyze loss function properties, introducing the enhanced counting loss to replace mean square counting loss. Furthermore, we transfer scale factor training in weight standardization to thresholds. Experiments show significant network performance improvements on datasets like CIFAR100, achieving state-of-the-art results on most others. While we've made notable progress, there's potential for further research into the optimal positive gradient ratio during training and the effectiveness of loss functions for SNN training in the time-based scheme. Additionally, extending our approach to ANNs, as demonstrated in [25], holds promise.

## 7 Acknowledgements

This work was supported by the National Natural Science Foundation of China under Grant No. 62176003 and No. 62088102.

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
