# Appendix for Exploring Loss Functions for Time-based Training Strategy in Spiking Neural Networks

**Yaoyu Zhu**
School of Computer Science
Peking University
Beijing, China 100871
yaoyu.zhu@pku.edu.cn

**Wei Fang**
School of Computer Science
Peking University
Beijing, China 100871
fwei@pku.edu.cn

**Tiejun Huang**
School of Computer Science
Peking University
Beijing, China 100871
tjhuang@pku.edu.cn

**Xiaodong Xie**
School of Computer Science
Peking University
Beijing, China 100871
donxie@pku.edu.cn

**Zhaofei Yu** [*]
Institute for Artificial Intelligence
Peking University
Beijing, China 100871
yuzf12@pku.edu.cn

## 1 Experimental Settings

The network architectures for each dataset are listed in Table. 1 in the main context, where the output layers are omitted since they are all fully-connected layers containing the same number of neurons as the number of classes in the dataset. The detailed training configuration (including data augmentation and hyper-parameters) is listed in Table. 1. In addition, the CIFARNet architecture in Table 1 in the main text is 128C3-256C3-P2-512C3-P2-1024C3-512C3-1024-512, while the VGG11 architecture is 128C3-128C3-P2-256C3-256C3-256C3-P2-512C3-512C3-512C3-P2-2048-2048. Here 128C3 means 128 channels of $3 \times 3$ convolution layer, and P2 means $2 \times 2$ pooling layer.

We conduct a binary search on threshold values in initialization to make sure that the average firing rate of each layer is around a certain number. In addition, we use supervisory signals in the output layer to guarantee that there will not exist a class that cannot be classified due to the dead output neuron problem. We infer and train the network in discrete time steps to better utilize the existing deep-learning frameworks. However, our network can be converted to continuous time seamlessly if we change the input encoding to whatever spike-based encoding.

In the implementation, directly using Eq. 20 and 21 to train the threshold may lead to negative thresholds. Therefore, we choose another approach to train the threshold: In each iteration, we first update parameters $\beta$ and $\gamma$ and then reset $\gamma$ to $\gamma_0$ while dividing $\beta$ and $\theta$ by $\frac{\gamma}{\gamma_0}$. To stabilize training (especially in the beginning), we scale up the gradients on output neurons corresponding to labels. The scaling factor is determined by the ratio of fired spike number between label neurons and other neurons. We use direct coding for static images as our network input (which is consistent with [4, 5]).

## 2 Proof of Theorem 4.1

**Theorem 4.1** *For a rate-based loss function $L(\boldsymbol{s})$ which does not involve spike timing, with a spike-to-timing derivative $\frac{\partial \boldsymbol{s}}{\partial \boldsymbol{t(s)}} = diag(\frac{\partial \int s_i(t_k)}{\partial t_k(s_i)}) = I$ where $(s_i, t_k)$ takes all neurons and spikes, it has an equivalent form $\mathcal{L}(\boldsymbol{t(s)}) = \sum_i \sum_{t_k(s_i)} \frac{\partial L}{\partial s_i(t_k)} \cdot t_k(s_i)$ defined on the timing of spikes.*

---

[*]Corresponding author

37th Conference on Neural Information Processing Systems (NeurIPS 2023).

Table 1: Experimental configurations

|  | MNIST | Fashion-MNIST | NMNIST | CIFAR10 | CIFAR100 |
|---|---|---|---|---|---|
| Optimizer | AdamW | AdamW | AdamW | SGD | SGD |
| Learning rate | 0.0005 | 0.0005 | 0.0005 | 0.0001 | 0.0001 |
| T (Time steps) | 5 | 5 | 30 | 12 | 16 |
| Weight decay | 0.001 | 0.002 | 0.005 | 0.0005 | 0.002 |
| $\text{Target}_{label}$[a] | 5 | 5 | 15 | 10 | 15 |
| $\text{Target}_{others}$[a] | 1 | 1 | 2 | 1 | 1 |
| Data augmentation | Norm[b] | Norm+HFlip[c] | - | Crop[d]+HFlip+AutoAug[e] | Crop+HFlip |

[a] We use the enhanced counting loss for all datasets. $\text{Target}_{label}$ and $\text{Target}_{others}$ are the target number of output spikes for label output neurons and other output neurons respectively.
[b] Norm: Input normalization.
[c] HFlip: Random horizontal flip.
[d] Crop: Random crop. For the CIFAR10 and CIFAR100 datasets, we both use an offset of $4$ in the random crop.
[e] AutoAug: Auto augment [1].

*Proof.* For a certain output spike $s_i(t_k)$, the derivative of the rate-based loss function $L$ to its timing $t_k(s_i)$

$$\frac{\partial L}{\partial t_k(s_i)} = \frac{\partial L}{\partial \int s_i(t_k)} \cdot \frac{\partial \int s_i(t_k)}{\partial t_k(s_i)} = \frac{\partial L}{\partial \int s_i(t_k)}$$

according to Eq. (8). If we want a time-based loss $\mathcal{L}$ to have the same derivative, then $\frac{\partial \mathcal{L}}{\partial t_k(s_i)} = \frac{\partial L}{\partial t_k(s_i)} = \frac{\partial L}{\partial \int s_i(t_k)}$. Therefore, if we take a single output spike timing $t_k(s_i)$ as the independent variable, we have

$$\mathcal{L}(t_k(s_i)) = \int \frac{\partial \mathcal{L}}{\partial t_k(s_i)} dt_k(s_i) = \int \frac{\partial L}{\partial \int s_i(t_k)} dt_k(s_i) = \frac{\partial L}{\partial \int s_i(t_k)} \cdot t_k(s_i) + C. \tag{1}$$

Here $C$ denotes a constant. The last equation is because the loss function $L$ does not involve spike timing in its expression, so $\frac{\partial L}{\partial \int s_i(t_k)}$ is independent of $t_k(s_i)$. In addition, the $\frac{\partial L}{\partial \int s_i(t_k)}$ is unrelated to other spike timings due to the same reason, so we can add equations of each output spike together and get $\mathcal{L}(\boldsymbol{t(s)}) = \sum_i \sum_{t_k(s_i)} \frac{\partial L}{\partial \int s_i(t_k)} \cdot t_k(s_i) + C$. Let $C = 0$ in this formula this theorem is proved. $\qquad\square$

## 3 Equivalent Time-based Loss Function for The Counting Loss

The counting loss has the form

$$L(\boldsymbol{s}, \textbf{target}) = \lambda \sum_{i=1}^{C} \left( \left( \int_0^T s_i(t)dt \right) - \text{target}_i \right)^2, \tag{2}$$

where $\lambda$ is a scaling constant, $C$ is the number of classes (also the number of output neurons), $\boldsymbol{s}$ is the spike train emitted by output neurons, and $s_i(t)$ denotes the output of the $i$-th neuron at time $t$.

As stated in the main text, we take

$$\frac{\partial \int s_i(t_k)}{\partial t_k(s_i)} = 1 \tag{3}$$

here to eliminate one degree of freedom. In this equation, $\int s_i(t_k)$ means the integral of $s_i(t_k)$ in a sufficiently small interval (which equals 1) when neuron $i$ fires at time $t_k$.

Suppose this neuron fires $m$ spikes ($m = \int_0^T s_i(t)dt$) and the target spike count for neuron $i$ is $\text{target}_i$. Then the derivative of the loss function to a spike it emits is

$$
\begin{aligned}
\frac{\partial L}{\partial \int s_i(t_k)} &= \lambda \frac{\partial \left( \int_0^T s_i(t)dt - \text{target}_i \right)^2}{\partial \int s_i(t_k)} = \lambda \frac{\partial \left( \sum_{k'=1}^m (\int s_i(t_k')) - \text{target}_i \right)^2}{\partial \int s_i(t_k)} \\
&= \lambda \frac{\partial \left( \sum_{k'=1}^m (\int s_i(t_k')) - \text{target}_i \right)^2}{\partial \sum_{k'=1}^m (\int s_i(t_k'))} \cdot \frac{\partial \sum_{k'=1}^m (\int s_i(t_k'))}{\partial \int s_i(t_k)} \\
&= \lambda \frac{\partial \left( m - \text{target}_i \right)^2}{\partial m} \cdot 1 = 2\lambda(m - \text{target}_i).
\end{aligned}
\tag{4}
$$

According to Theorem 4.1, the form of the equivalent time-based loss function $\mathcal{L}$ is:

$$
\mathcal{L}(\boldsymbol{t}(\boldsymbol{s})) = \sum_i \sum_{t_k(s_i)} \frac{\partial L}{\partial s_i(t_k)} \cdot t_k(s_i) = \sum_i \sum_{t_k(s_i)} 2\lambda \left( \int_0^T s_i(t)dt - \text{target}_i \right) \cdot t_k(s_i).
\tag{5}
$$

**Total Derivatives for a Neuron in Counting Loss:**

According to Eq. (4), the total gradients for the $m$ spikes fired by one neuron is

$$
m \cdot \frac{\partial L}{\partial t_k(s_i)} = 2\lambda m(\text{target}_i - m).
\tag{6}
$$

Note that typically $\text{target}_{label} \geq m \geq \text{target}_{others}$ (in practice, we clip the gradients to 0 if label output neurons fire more than desired or other output neurons fire fewer than desired). As a result, the total gradient scale for a label output neuron first increases and then decreases quadratically as the number of fired spikes grows. For other neurons, the gradient scale for other neurons increases quadratically as the number of fired spikes grows.

## 4 Derivation of The Enhanced Counting Loss

To guarantee the total gradients for $m$ spikes fired by one neuron to be $2\lambda(\text{target}_i - m)$, we need to set $\frac{\partial L}{\partial \int s_i(t_k)} = 2\lambda \frac{(\text{target}_i - m)}{m}$. Compared with Eq. 4, we just need to turn the equation in the second last line (deduction of other parts are unchanged)

$$
\frac{\partial L(m)}{\partial m} = \frac{\partial \left( m - \text{target}_i \right)^2}{\partial m} = 2(m - \text{target}_i)
\tag{7}
$$

into

$$
\frac{\partial L(m)}{\partial m} = 2\frac{(m - \text{target}_i)}{m}.
\tag{8}
$$

As a result,

$$
L(m) = \int 2\frac{(m - \text{target}_i)}{m} dm = 2 \int \left( 1 - \frac{\text{target}_i}{m} \right) dm = 2(m - \text{target}_i \ln m).
\tag{9}
$$

Since $\ln m$ is undefined when $m \leq 0$, we can set (for instance) $L(m) = 0$ when $m \leq 0$. In this case, the formal form of enhanced counting loss is

$$
L(\boldsymbol{s}, \textbf{target}) = 2\lambda \sum_{i=1}^C \left( \left( \int_0^T s_i(t)dt \right) - \text{target}_i \ln \left( \int_0^T s_i(t)dt \right) \right).
\tag{10}
$$

with approximation

$$
\frac{\partial \int s_i(t_k)}{\partial t_k(s_i)} = -1.
\tag{11}
$$

Table 2: Expression of different losses

| Name | Expression | Rate to Timing Approximation |
|---|---|---|
| Counting (MSE) | $\lambda \sum_{i=1}^{C} \left( \left( \int_0^T s_i(t)dt \right) - \text{target}_i \right)^2$ | $\frac{\partial \int s_i(t_k)}{\partial t_k(s_i)} = -1$ |
| Spike train (kernel) | $\lambda \sum_{i=1}^{C} \int_0^T \left( \kappa * (s_i - s_i^{target})(t) \right)^2 dt$ | $\frac{\partial \int s_i(t_k)}{\partial t_k(s_i)} = -1$ |
| Temporal efficient training | $-\lambda \sum_{t=1}^{T} \left( \sum_{i=1}^{C} \log \frac{\exp(s_i[t])}{\sum_{j=1}^{C} \exp(s_j[t])} \cdot s_i^{target}[t] \right)$ | $\frac{\partial \int s_i(t_k)}{\partial t_k(s_i)} = -1$ |
| Spike train (timing) | $\lambda \sum_{i=1}^{C} \left( \sum_{j=1}^{n_i} \left( t_j(s_i) - t_j^{target}(s_i) \right) \right)^2$ | - |
| Time to first spike (CE) | $-\frac{\lambda}{\beta} \sum_{i=1}^{C} \text{target}_i \log \frac{\exp(-\beta t_1(s_i))}{\sum_{i=1}^{C} \exp(-\beta t_1(s_i))}$ | - |

# 5 Additional Experiments

## 5.1 Experiments on Different Losses

In addition to the main body of our paper, we have also tried several other losses on the Fashion-MNIST dataset, as listed in Table. 2. The reason we choose the Fashion-MNIST dataset is that this dataset is relatively hard among those requiring not-so-much time to train. The network structure we use here follows that in the main body of our paper.

The MSE and CE in the table mean mean-square-error and cross-entropy, respectively. For the temporal efficient training (TET) loss [2], we use [] instead of () since it is defined in the discrete time steps. The $s_i^{\text{target}}[t]$ is 1 for label output neurons and 0 for other neurons on each time step.

For the spike train (timing) loss, we set $T$ target spikes with firing time $t_j^{\text{target}}(s_i) = 0.5$ to $T - 0.5$ for label output neurons. For other output neurons, we set the target spikes for certain amounts of time after the actual firing time dynamically, where the time difference between target spikes and actual output spikes is set to $\frac{T}{\sqrt{C}}$ (where $C$ is the number of classes in the dataset) empirically to balance the gradients.

Table 3: Results of different gradients on Fashion-MNIST

| Loss function | Test Accuracy |
|---|---|
| Enhanced counting (MSE) | 94.03% |
| Counting (MSE) | 93.72% |
| Spike train (kernel) | 93.81% |
| Temporal efficient training | 80.91% |
| Spike train (timing) | 94.03% |
| Time to first spike (CE) | 91.89% |

For the time-to-first-spike loss, we empirically select scaling factor $\beta = \frac{1}{T}$ (to reduce the gradient vanishing problem). In addition, $t_1(s_i)$ means the timing of the first spike a neuron fires and we set $t_{unfired} = 4T$, where $t_{unfired}$ denotes the $t_1(s_i)$ when a neuron does not emit any spikes at all.

The results are shown in Table. 3 and Figure. 1. From the results, we can see that counting loss, spike train (kernel) loss, and spike train (timing) loss can all train the network with fair accuracy. The TTFS loss behaves not as well as the three above losses, while the TET loss [2] behaves much worse. Actually, if the $t_{unfired}$ is not selected carefully, its performance will also be degraded to the TET loss. This effect will be discussed in detail in the next section.

From Figure. 1, we can see that the counting loss and spike train (timing) loss have a larger overall loss scale. Oppositely, TTFS loss and TET loss have a smaller loss scale. For the ratio of positive gradients, we can see that the TET loss has the most imbalanced ratio, even in the early stages. This is an important reason that it cannot train the network well. The ratio of positive gradients for the TTFS loss also surpasses the other three losses which can train the network well.

Although the spike train (timing) loss performs as well as the enhanced counting loss for the Fashion-MNIST dataset, it is currently more challenging to tune on larger networks, resulting in a CIFAR10 dataset accuracy of only 90.09%. In addition, the loss function almost does not affect the time and space complexity of the overall algorithm. The reason is that it only relates to the last layer of the

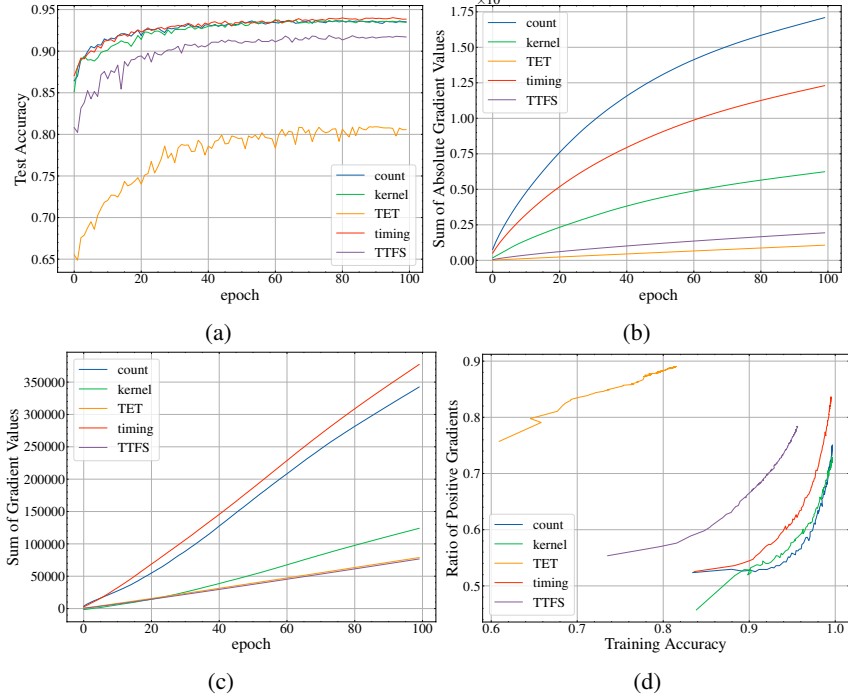

Figure 1: Results of different losses. (a) Test accuracy along with training epoch index. (b) Sum of absolute gradient values along with training epoch index. (c) Sum of gradient values along with training epoch index. (d) The ratio of positive gradients along with training accuracy.

network (with size $T \times N \times C$, where $T$ is simulation time, $N$ is the batch size, $C$ is the number of classes in the dataset), which occupies only a tiny proportion of calculation and memory. In addition, the loss function is unnecessary for inference. As a result, the enhanced counting loss proves to be more effective in terms of parameter tuning and overall performance.

## 5.2 Parameter Selection For TTFS loss

In this section, we show the influence of parameter selection (especially $t_{unfired}$) on the performance of the TTFS (CE) loss.

Before going into the experiments, we first derive a property of the TTFS (CE) loss and verify a statement in the main text. In section 4.3 of the main body of our paper, we wrote that the sum of gradients on spike timing should be positive. Actually, we can construct a loss function that has almost zero-summed gradients by setting $t_{unfired}$ for the TTFS (CE) loss.

**A loss resulting in zero-summed gradients.** Firstly, we can rewrite the TTFS CE loss:

$$L(\boldsymbol{s}, \textbf{target}) = -\frac{\lambda}{\beta} \sum_{i=1}^{C} \text{target}_i \log \frac{\exp(-\beta t_1(s_i))}{\sum_{i=1}^{C} \exp(-\beta t_1(s_i))}$$
$$= -\frac{\lambda}{\beta} \text{target}_{label} \log \frac{\exp(-\beta t_1(s_{label}))}{\sum_{i=1}^{C} \exp(-\beta t_1(s_i))}, \quad (12)$$

When the neuron fires a spike, the gradient of the timing of this spike to the loss function is

$$\frac{\partial L}{\partial t_1(s_i)} = \text{target}_i - \frac{\exp(-\beta t_1(s_i))}{\sum_{j=1}^{C} \exp(-\beta t_1(s_j))}. \quad (13)$$

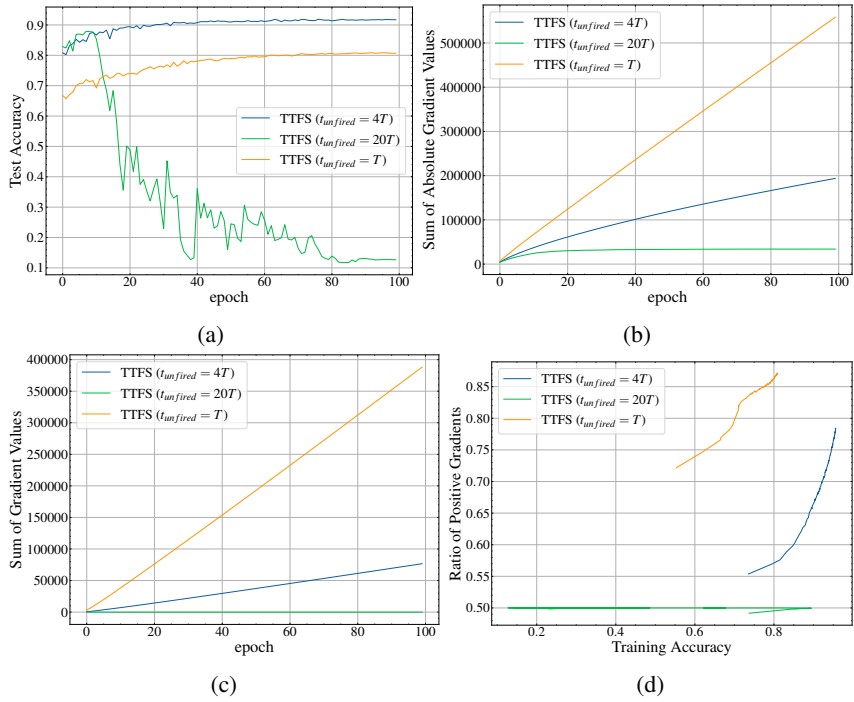

Figure 2: Results of different parameters. (a) Test accuracy along with training epoch index. (b) Sum of absolute gradient values along with training epoch index. (c) Sum of gradient values along with training epoch index. (d) The ratio of positive gradients along with training accuracy.

The TTFS CE loss has a nice property when all output neurons fire spikes:

$$
\begin{aligned}
\sum_{i=1}^{C} \frac{\partial L}{\partial t_1(s_i)} &= \sum_{i=1}^{C} \left( \text{target}_i - \frac{\exp(-\beta t_1(s_i))}{\sum_{j=1}^{C} \exp(-\beta t_1(s_j))} \right) \\
&= 1 - \frac{\sum_{i=1}^{C} \exp(-\beta t_1(s_i))}{\sum_{j=1}^{C} \exp(-\beta t_1(s_j))} = 0,
\end{aligned}
\tag{14}
$$

which means the sum of gradients on output spikes is naturally summed to zero.

However, there are some output neurons that will not fire spikes and cannot receive this gradient, which will break this property. To deal with this problem, we just need to set $t_{unfired}$ to be a very large number when a neuron does not fire any spikes for a certain sample so that $\exp(-\beta t_{unfired}) \to 0$. In addition, we make use of the supervisory signal to force the neuron corresponding to the label to fire at least once during the forward stage. Then if we denote $f_i$ as whether output neuron $i$ fires a spike at any time (1 for fire and 0 for not, note $f_{label}$ must be 1 due to supervisory signal), the sum of

gradients in the output layer is

$$\sum_{i=1}^{C} \frac{\partial L}{\partial t_1(s_i)} = \sum_{i=1}^{C} f_i \left( \text{target}_i - \frac{\exp(-\beta t_1(s_i))}{\sum_{j=1}^{C} \exp(-\beta t_1(s_j))} \right)$$

$$= \sum_{i=1}^{C} f_i \text{target}_i - \frac{\sum_{i=1}^{C} f_i \exp(-\beta t_1(s_i))}{\sum_{j=1}^{C} \exp(-\beta t_1(s_j))}$$

$$= 1 - \left( 1 - \frac{\sum_{i=1}^{C} (1-f_i) \exp(-\beta t_1(s_i))}{\sum_{j=1}^{C} \exp(-\beta t_1(s_j))} \right)$$

$$= \frac{\sum_{i=1}^{C} (1-f_i) \exp(-\beta t_1(s_i))}{\sum_{j=1}^{C} \exp(-\beta t_1(s_j))}$$

$$\approx \frac{0}{\sum_{j=1}^{C} \exp(-\beta t_1(s_j))} = 0. \tag{15}$$

The approximation in line 5 is because $\exp(-\beta t_1(s_i)) \to 0$ for neurons that do not fire. In line 6, note that the label output neuron always fires, therefore the denominator is bigger than 0.

**Experiments.** Here we take three options for the parameter $t_{unfired}$. The first is $t_{unfired} = T$, denoting a situation with too many overall positive gradients. The second is $t_{unfired} = 4T$, denoting a circumstance with adequate overall positive gradients. The third is $t_{unfired} = 20T$, denoting a situation with zero-summed overall gradients.

The results are shown in Table. 4 and Figure. 2. We can see that when $t_{unfired} = T$, the output layer receives too many (positive) gradients and the ratio of positive gradients is too large. The network cannot converge well with this loss function. On the other hand, when $t_{unfired} = T$, the test accuracy increases in the first few epochs and then drops drastically. We observe an increase in threshold simultaneously. Note that due to weight standardization, the threshold is a primary influence factor of the firing rate.

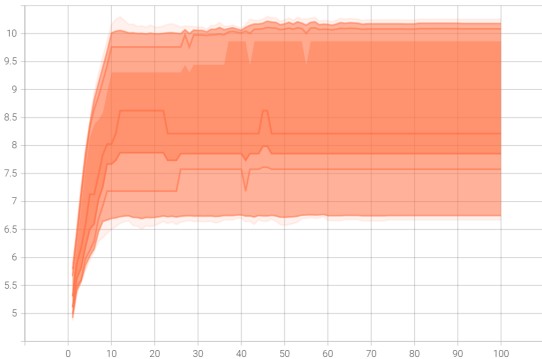

Figure 3: Threshold distribution of the last layer

## 5.3  Experiments on Inference Time

We want to show that our time-based training algorithm can lead to early-stage inference in this section. To test the time for inference, we will change the prediction strategy of our network in the following experiments. Specifically, we will use the early-exit mechanism to determine the inference time. We will simulate our network in discrete time steps and decide the prediction result and inference time by selecting the output neuron that first outputs a spike. If two or more output neurons fire spikes at the

Table 4: Results of different gradients between layers

| $t_{unfired}$ | Test Accuracy |
|---|---|
| $T$ | 80.85% |
| $4T$ | 91.89% |
| $20T$ | 87.90% |

Table 5: Accuracy and inference time

| Epoch | 10 | 20 | 30 | 40 | 50 | 60 | 70 | 80 | 90 | 100 |
|---|---|---|---|---|---|---|---|---|---|---|
| Test Accuracy | 98.83 | 98.86 | 98.99 | 99.04 | 99.04 | 99.09 | 99.09 | 99.09 | 99.09 | 99.09 |
| Inference time | 0.189 | 0.169 | 0.159 | 0.147 | 0.147 | 0.137 | 0.129 | 0.128 | 0.128 | 0.128 |

same time step, we will wait until a time step at which
they differ. If the prediction cannot be determined until
the end of the simulation or the prediction is wrong, we will count the inference time as $T$. We expect
our algorithm can show a performance increase with comparable inference time compared with the
SNN trained by [3].

We have conducted experiments on the MNIST dataset to compare with [3]. We converted the input
into spikes conforming to [3] (with a total time step of $T = 10$). We trained our algorithm for 100
epochs, and the test accuracy and inference time are shown in the following table (note that we list
the accumulated best accuracy and accumulated fastest time).

Compared to Figure 9 in [3], our algorithm has an advantage in SNN testing accuracy and comparable
inference time.