# OpenReview forum: "Exploring Loss Functions for Time-based Training Strategy in Spiking Neural Networks"
_NeurIPS.cc/2023/Conference — NeurIPS 2023 spotlight_

### Official Review · Reviewer_jtny · 2023-07-02

**Soundness:** 3 good
**Presentation:** 3 good
**Contribution:** 3 good
**Rating:** 7
**Confidence:** 5

**Summary:**

This paper focuses on the loss functions for time-based training schemes of SNNs, which propagate gradients only when the neurons fire a spike. The authors propose to map rate-based loss functions to time-based ones and explain why they also work. Besides, the authors propose the enhanced counting loss to replace the commonly used mean square counting loss. The experimental results show that the proposed method achieves SOTA performance among time-based methods.

**Strengths:**

1. The training algorithm studied in this paper is relatively novel and fits the event-driven feature of SNN. It also provides a novel aspect to link temporal coding and rate coding in SNNs.

2. This paper provides a solid analysis of applying rate-coded losses in time-based training schemes. The proposed enhanced counting loss reasonably stabilizes the whole training process and thus improves performance.

3. This paper is relatively well-organized and well-written.

**Weaknesses:**

The training algorithm studied in this paper cannot achieve comparable performance compared with BPTT-based methods.

**Questions:**

1. Have the authors tried loss functions other than counting-based losses? For example, loss functions defined on spike timings as in [1] or time-to-first-spike based loss function as in [2]?

2. How is the simulation performed? Is it simulated in continuous time or discrete time steps? If it is simulated discretely, what time steps do you use in your experiment?

[1]Bohte, S. M., Kok, J. N., & La Poutre, H. (2002). Error-backpropagation in temporally encoded networks of spiking neurons. Neurocomputing, 48(1-4), 17-37.
[2]Mostafa, H. (2017). Supervised learning based on temporal coding in spiking neural networks. IEEE Transactions on neural networks and learning systems, 29(7), 3227-3235.

**Limitations:**

There’s no potential negative societal impact that should be addressed.

---

> ### Author Rebuttal · Authors · 2023-08-08
>
> Thank you for acknowledging our research direction and the innovative approach we have employed. We are truly grateful for your recognition of the solid analysis, well-structured organization, and clear exposition in our work. In response to your valuable feedback, we are fully committed to addressing any concerns you have raised and providing comprehensive responses to the questions, as detailed in the following sections.
>
>
> > The training algorithm studied in this paper cannot achieve comparable performance compared with BPTT-based methods.
>
> We appreciate your attention to this matter. Our proposed method differs from BPTT-based approaches in two significant aspects.
> Firstly, we calculate gradients of spike timings generated by the loss function, whereas BPTT-based methods compute gradients of the 'spike scale' (as depicted in Figure 1(b)(d)).
> Secondly, our method operates in an event-driven manner, wherein information propagates solely through spikes during both forward and backward propagation. In contrast, BPTT-based techniques propagate gradient information even in the absence of emitted spikes, as exemplified by the surrogate gradient's approximation of $\frac{\partial s}{\partial u}$ regardless of spike occurrence.
>
> The event-driven property of our method poses a challenge during training compared to BPTT-based approaches due to the sparse gradient propagation path. Additionally, as our learning scheme is relatively novel when compared with BPTT-based methods, its performance and convergence speed have not yet surpassed those of the latter.
> However, the event-driven property empowers our learning scheme with enhanced biological plausibility and the potential for more efficient optimization when executed on neuromorphic hardware.
>
>
> > Have the authors tried loss functions other than counting-based losses? For example, loss functions defined on spike timings as in [1] or time-to-first-spike based loss function as in [2]?
>
> We have tried other loss functions, including these two loss functions. The corresponding results are listed in Section 5.1 in the supplementary material.
> We have conducted the experiments on the Fashion-MNIST dataset, and the results are listed in Table.3 in the appendix.
> From the results, cross-entropy losses (whether with respect to firing rate such as the temporal efficient training loss [3] or firing time such as the time to first spike loss(CE)) cannot achieve good results on this dataset. This is partly due to the abnormal ratio of positive gradients (Figure.1(d) and Figure.2(d) in the appendix). The ratio is easily to be too large.
> For the spike train (kernel) and spike train (timing) loss, they behave normally in the Fashion-MNIST dataset. However, when extending to larger datasets like CIFAR-10, the spike train (timing) loss cannot achieve a good result (90.09\% compared with 93.54\% for the enhanced counting loss).
> Therefore, currently the enhanced counting loss is the best one under our attempt among different losses for the time-based SNN training scheme.
>
>     [1]Bohte, S. M., Kok, J. N., & La Poutre, H. (2002). Error-backpropagation in temporally encoded networks of spiking neurons. Neurocomputing, 48(1-4), 17-37.
>     [2]Mostafa, H. (2017). Supervised learning based on temporal coding in spiking neural networks. IEEE Transactions on neural networks and learning systems, 29(7), 3227-3235.
>     [3]Deng, S., Li, Y., Zhang, S., & Gu, S. (2021, October). Temporal Efficient Training of Spiking Neural Network via Gradient Re-weighting. In International Conference on Learning Representations.
>
>
> > How is the simulation performed? Is it simulated in continuous time or discrete time steps? If it is simulated discretely, what time steps do you use in your experiment?
>
> We would like to emphasize that our training approach does not require splitting the entire process into discrete time steps. As shown in Figure 1, we only need to record information at spike times (such as the precise timing and slope of the membrane potential) to train our network, and there is no inherent need to use clock-driven methods. However, to better integrate with current deep learning frameworks, we convert the training process from continuous time to discrete time steps in our simulation. For MNIST and Fashion-MNIST, we set the number of time steps to 5, for CIFAR10 we set it to 12, for CIFAR100 we set it to 16, and for N-MNIST we set it to 30 (these parameters are consistent with [4]). For further information about hyper-parameters, please refer to Table S1 in the appendix.
>
>     [4] Zhu, Y., Yu, Z., Fang, W., Xie, X., Huang, T., & Masquelier, T. (2022). Training Spiking Neural Networks with Event-driven Backpropagation. In 36th Conference on Neural Information Processing Systems (NeurIPS 2022).

---

> > ### Comment · Reviewer_jtny · 2023-08-11
> > **Thanks for the response**
> >
> > As my concerns are well-addressed. I would like to raise my score to 7.

---

### Official Review · Reviewer_98Db · 2023-07-02

**Soundness:** 3 good
**Presentation:** 3 good
**Contribution:** 3 good
**Rating:** 8
**Confidence:** 5

**Summary:**

This paper explores the loss function for SNNs and proves that rate-based loss functions can be also used in the time-based training scheme. Besides, the authors propose a new loss function called enhanced counting loss, which improves the network performance (compared with previously-used mean square counting loss) by providing adequate positive overall gradients for time-based training schemes in SNN training. In addition, they propose a new normalization approach, which helps the training process by tuning the threshold instead of standardizing the weights. In summary, this paper brings some interesting ideas to the community of SNNs.

**Strengths:**

1. This paper focuses on time-based learning of SNNs, which is a very important direction of SNNs as it can better utilize the temporal information of SNNs. There is little work at present. The authors propose some interesting ideas (loss function, normalization approach), which is very impressive.
2. The proposed framework can explain why the derivative of integration of spikes with respect to the firing time is set to -1 in many previous studies.
3. The paper is well-written and includes rigorous proofs.
4. The proposed approach outperforms previous time-based training methods.

**Weaknesses:**

I think the description from Eqs.(1)-(5) is hard for readers unfamiliar with time-based learning to understand. They need to check the reference [58]. I suggest adding the detailed derivation in Appendix.

**Questions:**

1. Despite better utilizing the temporal information of SNNs, what are the advantages of time-based learning methods compared to activation-based approaches? Could the authors give some comments?
2. I would like to see some comments on the time-based learning of SNNs on neuromorphic chips. Is it more energy efficient?
3. The loss function (13) seems not well-defined when no spikes are emitted (the value in the ln() function is 0). I would like to see some explanations to cover this corner case.

**Limitations:**

N.A.

---

> ### Author Rebuttal · Authors · 2023-08-08
>
> Thank you for providing such an encouraging and positive response. We sincerely appreciate your recognition of the significance of our work, its alignment with an important direction, its well-written nature, and the inclusion of rigorous proofs. In light of your feedback, we are fully dedicated to addressing any concerns you have raised and offering comprehensive responses to the questions, as delineated in the subsequent sections.
>
> > I think the description from Eqs.(1)-(5) is hard for readers unfamiliar with time-based learning to understand. They need to check the reference [58]. I suggest adding the detailed derivation in Appendix.
>
> Thank you for your valuable suggestion. We will add the corresponding derivation as well as an algorithm description to our paper (please refer to the response to Reviewer LeWP for the algorithm description).
>
> > Despite better utilizing the temporal information of SNNs, what are the advantages of time-based learning methods compared to activation-based approaches? Could the authors give some comments?
>
> The key point is that our approach is event-driven.
> One can compare Figure.1(a)(b) and (c)(d) to understand the difference between the time-based training scheme and the activation-based training scheme:
> In Figure.1(a)(b), the forward and backward propagation of activation-based training schemes cannot be simulated in continuous time, since the surrogate gradient in the backward stage is also propagated through neurons that are not fired.
> In Figure.1(c)(d), the simulation of time-based training schemes can operate in continuous time, since in both forward and backward stage the information propagation between neurons are conveyed by spikes.
> This event-driven property guarantees the possibility of network simulation in continuous time, has the potential to save energy on neuromorphic chips, and provides more biological plausibility.
>
>
> > I would like to see some comments on the time-based learning of SNNs on neuromorphic chips. Is it more energy efficient?
>
> Our learning scheme has the potential to exhibit enhanced energy efficiency when executed on neuromorphic hardware.
>
> To highlight the hardware efficiency advantage, we conduct a comparison of the number of operations between our method and Backpropagation Through Time (BPTT) algorithms when dealing with sparse spikes. Although we analyze a single fully-connected layer for simplicity, the analysis extends similarly to other types of layers and the entire network.
>
> During training, BPTT algorithms necessitate unfolding through the time axis, as depicted in Figure 1(a)(b). Consequently, this leads to a corresponding number of operations of at least $O(TC_{in}C_{out})$, where $T$ denotes the total time steps, and $C_{in}$ and $C_{out}$ represent the number of input and output neurons, respectively. In contrast, temporal learning algorithms, such as ours, only need to address scenarios where a specific neuron fires a spike and records pertinent information. During the forward stage, a spike fired by an input neuron influences the state of both itself and all output neurons, culminating in a total of $O(C_{out})$ operations. During the backward stage, a spike discharged by an output neuron requires propagating gradient information to all intervening spikes between this spike and the last spike fired by the same neuron (thus, all input spikes are processed once during this stage). Therefore, by denoting the average firing rate of input and output neurons as $\alpha$ and $\beta$, the number of operations for this layer is determined as $O(T(\alpha C_{in}C_{out}+\beta C_{in}))$.
>
> In cases of sparse spikes, event-based learning algorithms exhibit an advantage, as $\alpha+\beta \ll 1$ under such circumstances.
>
>
> > The loss function (13) seems not well-defined when no spikes are emitted (the value in the ln() function is 0). I would like to see some explanations to cover this corner case.
>
> Thank you for pointing it out. The integral $\int_0^T s_i(t) dt$ must be an integer since it represents the number of spikes fired by neuron $i$ during simulation time ($0$ to $T$), so only the derivation at integer point is meaningful. As a result, we can let $f(x) = x - \text{target}_i \ln(x)$ when $x>0$ and $f(x) = 0$ when $x=0$ to fix this bug. Then the loss function is
> $$L( \boldsymbol{s}, \boldsymbol{target} ) = 2\lambda \sum\_{i=1}^C f \left( \int_0^T s_i(t) dt \right).$$

---

> > ### Comment · Reviewer_98Db · 2023-08-12
> >
> > I have read the point-to-point responses to my questions, which can address my concerns.

---

### Official Review · Reviewer_LeWP · 2023-07-04

**Soundness:** 4 excellent
**Presentation:** 3 good
**Contribution:** 3 good
**Rating:** 6
**Confidence:** 4

**Summary:**

In this work, the authors propose a framework that applies the rate-coding-based loss functions to time-based training. They show that the proposed method outperforms existing time-based ones.

**Strengths:**

1. The proposed method suggests a way to combine different approaches to train spiking neural networks.
2. It provides detailed derivation and a clear explanation of the pipeline.

**Weaknesses:**

1. Overall, the performance of time-based models suffers a deficit compared to other SNN methods. What kind of evaluable benefit does this kind of training bring to the current paper?

2. What's the difference between encoding the error time w.r.t to the membrane potential and the spike time?

3. It would be better to have an Algorithm description in the main context.

4. It lacks a clear demonstration of what kind of new information/feature is captured so that the overall performance is improved.

**Questions:**

Please check the weakness.

---

> ### Author Rebuttal · Authors · 2023-08-08
>
> We express our sincere gratitude for providing constructive and insightful feedback such as the addition of the algorithm description. It is truly gratifying to know that you value the meticulous derivation and lucid explanation of our pipeline. In light of your input, we are fully dedicated to addressing your concerns as outlined in the forthcoming sections.
>
> > Overall, the performance of time-based models suffers a deficit compared to other SNN methods. What kind of evaluable benefit does this kind of training bring to the current paper?
>
> The primary objective of our research is to develop a fully biologically plausible learning scheme akin to STDP. These approaches possess key properties, including being event-driven, online, and local. While more biologically plausible schemes may incur performance trade-offs, they hold significant value in explaining learning rules observed in biological brains.
>
> Our work focuses on event-driven property. One can gain insight into the distinction between time-based and activation-based training schemes by comparing Figure 1(a)(b) and (c)(d):
> In Figure 1(a)(b), the forward and backward propagation of activation-based training schemes cannot be simulated in continuous time, as the surrogate gradient in the backward stage is also propagated through neurons that have not fired.
> Meanwhile, in Figure 1(c)(d), the simulation of time-based training schemes can indeed operate in continuous time, as both the forward and backward stages entail information propagation between neurons conveyed by spikes.
>
> Apart from conferring more biological plausibility, this event-driven property ensures the feasibility of network simulation in continuous time and holds the potential to conserve energy when executed on neuromorphic chips.
>
> > What's the difference between encoding the error time w.r.t to the membrane potential and the spike time?
>
> The membrane potential and spike time are not against each other since they pertain to distinct stages in SNN time-based training. You can refer to Figure.1: Suppose there are $N$ layers in total, then in time-based training, the gradient propagation path is loss -> (spike timing in layer $N$ -> membrane potential in layer $N$) -> (spike timing in layer $N-1$ -> membrane potential in layer $N-1$) -> ... -> (spike timing in layer $1$ -> membrane potential in layer $1$), and there is an additional path from membrane potential in layer $n$ to weight between layer $n$ and $n-1$.
>
> > It would be better to have an Algorithm description in the main context.
>
> Although there exists a continuous version of our algorithm, we implement the discrete version since it aligns more seamlessly with existing deep learning frameworks, rendering it easier to implement and enabling better utilization of CUDA acceleration. The following is the discrete version of our algorithm and will be added to our paper:
> $$\begin{array}{l}
> \text{Input: Input spike train $\boldsymbol s^{(0)}$, weight for all layers $\boldsymbol W^{(n)}(n = 1, ..., N)$, total simulation time }T\\\\
> \text{Output:}\nabla\boldsymbol W^{(n)}(n=1,...,N)\\\\
> \text{//Forward}\\\\
> \text{Initialize$\boldsymbol{Neuron}^{(n)}(n=1,...,N)$: Neurons with states for all layers}\\\\
> \text{for $n=0$ to $N-1$ do}\\\\
> \qquad\text{for $t=0$ to $T-1$ do}\\\\
> \qquad\qquad\text{Update state of $\boldsymbol{Neuron}^{(n+1)}$by Eq.(1)}\\\\
> \qquad\qquad\text{if Membrane potential $\boldsymbol u^{(n+1)}\geq\theta$ then}\\\\
> \qquad\qquad\qquad\boldsymbol s^{(n+1)}=1\\\\
> \qquad\qquad\qquad\text{Reset state of }\boldsymbol{Neuron}^{(n+1)}\\\\
> \qquad\qquad\text{end if}\\\\
> \qquad\text{end for}\\\\
> \text{end for}\\\\
> \text{//Backward}\\\\
> \text{Initialize$\nabla\boldsymbol W^{(n)},\frac{\partial L}{\partial\boldsymbol t(s)^{n}}\gets0$for }n=1,2,...,N\\\\
> \text{Calculate$\frac{\partial L}{\partial\boldsymbol t(s)^{N}}$by Eq.(13)}\\\\
> \text{for $n=N$ to 1 do}\\\\
> \qquad\text{for $t=T-1$ to 0 do}\\\\
> \qquad\qquad\text{if $\boldsymbol s^{(n)}(t)=1$ then}\\\\
> \qquad\qquad\qquad\text{Change$\frac{\partial L}{\partial\boldsymbol u^{(n)}}$to$\frac{\partial L}{\partial \boldsymbol t(s^{(n)})}\frac{\partial\boldsymbol t(s^{(n)})}{\partial\boldsymbol u^{(n)}(t)}$by part of Eq.(3)(4)}\\\\
> \qquad\qquad\text{end if}\\\\
> \qquad\qquad\text{Add$\frac{\partial L}{\partial\boldsymbol u^{(n)}}\frac{\partial\boldsymbol u^{(n)}}{\partial\boldsymbol W^{(n)}}$to$\nabla\boldsymbol W^{(n)}$by Eq.(3)}\\\\
> \qquad\qquad\text{Add$\frac{\partial L}{\partial\boldsymbol u^{(n)}}\frac{\partial\boldsymbol u^{(n)}}{\partial\boldsymbol t(s^{(n-1)})}$to$\frac{\partial L}{\partial \boldsymbol t(s^{(n-1)})}$by Eq.(4)}\\\\
> \qquad\text{end for}\\\\
> \text{end for}\\\\
> \end{array}$$
>
> > It lacks a clear demonstration of what kind of new information/feature is captured so that the overall performance is improved.
>
> In section 4.3, we have analyzed the deficiency of counting loss: Suppose an output neuron corresponding to the label emits $m$ spikes, then the total gradients on these spikes are $m (\text{target}\_{label} - m)$ (Eq(12)). This is a concave quadratic function, which means firing one spike will receive a total gradient equal to firing $\text{target}\_{label} - 1$ spikes and less than firing $2$ to $\text{target}\_{label} - 2$ spikes. Recall that the total time-based gradient means the cumulative pushing of all spikes for how much earlier, remains constant across layers (Eq(6)). A quadratic function is not reasonable under this concern.
> Our proposed enhanced counting loss converts this function $m (\text{target}\_{label} - m)$ to a decreasing linear function $\text{target}\_{label} - m$, which is more reasonable regarding the total gradient.
>
> Therefore in experiments, we show the total gradient scale (both original and after absolute function), and the ratio of positive gradients w.r.t the training accuracy. Results show that our proposed method decreases the total gradient scale and makes the ratio of positive gradients more balanced.

---

> > ### Comment · Reviewer_LeWP · 2023-08-15
> > **Response to the rebuttal**
> >
> > Thanks for the rebuttal. It addresses my questions. I think my current score is suitable.

---

### Official Review · Reviewer_hbi5 · 2023-07-10

**Soundness:** 3 good
**Presentation:** 3 good
**Contribution:** 3 good
**Rating:** 6
**Confidence:** 4

**Summary:**

This paper focuses on the time-based training approach for spiking neural networks (SNNs). It first explains why rate-based loss functions can be used in time-based training for SNNs by establishing a link between rate-based losses and time-based ones. Then it does some analysis on the overall gradient provided by the loss function and proposes the enhanced counting loss based on this analysis. At last, it proposes a new normalization method that changes the threshold instead of the weight scale. Experimental results on classification tasks show improved performance

**Strengths:**

1. Exploring event-driven learning for SNNs is important.
2. This paper is technically solid in analyzing various aspects of rate-based loss functions.
3. The proposed enhanced counting loss and threshold training are shown to be effective by the experiments.


**Weaknesses:**

This paper has a relatively narrow scope, focusing solely on improving counting-based loss functions for time-based training schemas.

**Questions:**


1. Is section 4.1 and 4.2 related to the proposed methods in this paper? I am somewhat confused about the role of this part in this paper, and hard to follow its conclusion.
2. Authors do not list BPTT-based methods in experiments. They say BPTT-based methods incorporate information when neurons do not fire in training, which improves the final performance. Does it mean BPTT-based methods have a better performance than time-based methods?
3. Is time-based learning of SNN more suitable for neuromorphic chips than activation-based learning?


**Limitations:**

See above

---

> ### Author Rebuttal · Authors · 2023-08-08
>
> We sincerely appreciate your constructive and insightful feedback. It is truly heartening to learn that you acknowledge the sound analysis presented in our paper and find merit in our newly proposed loss function, which contributes to the stabilization of the entire training process. We are committed to addressing your concerns and providing comprehensive responses to the questions you have raised, as outlined in the following sections.
>
> > This paper has a relatively narrow scope, focusing solely on improving counting-based loss functions for time-based training schemas.
>
> Thank you for your valuable feedback. We wish to underscore that, alongside the counting-based loss function tailored for the time-based training schema, we have also introduced a mechanism to incorporate rate-based loss functions into the temporal training approach. Moreover, the potential to apply time-based loss functions to rate-based training schemes in a reverse manner offers a compelling avenue for future exploration.
>
> Furthermore, our method proposing the transfer of training scale factors utilized by weight standardization into thresholds is not restricted to specific SNN training algorithms. We believe that our work provides a foundation for future research in this area, and we genuinely appreciate your keen interest in our work.
>
>
> > Is section 4.1 and 4.2 related to the proposed methods in this paper? I am somewhat confused about the role of this part in this paper, and hard to follow its conclusion.
>
> Thank you for raising this point. Sections 4.1 and 4.2 of our paper furnish a theoretical basis for applying rate-based losses in temporal SNN training. The analysis presented in these sections elucidates the underlying principles behind both the counting loss and our proposed enhanced counting loss and elucidates why they are effective in our approach. We will clarify it in the revised version.
>
>
> > Is time-based learning of SNN more suitable for neuromorphic chips than activation-based learning?
>
> Yes. The event-driven nature of our learning scheme contributes to its increased biological plausibility and suitability for continuous-time learning while also exhibiting enhanced efficiency when executed on neuromorphic hardware.
>
> To highlight the hardware efficiency advantage, we conduct a comparison of the number of operations between our method and Backpropagation Through Time (BPTT) algorithms when dealing with sparse spikes. Although we analyze a single fully-connected layer for simplicity, the analysis extends similarly to other types of layers and the entire network.
>
> During training, BPTT algorithms necessitate unfolding through the time axis, as depicted in Figure 1(a)(b). Consequently, this leads to a corresponding number of operations of at least $O(TC_{in}C_{out})$, where $T$ denotes the total time steps, and $C_{in}$ and $C_{out}$ represent the number of input and output neurons, respectively. In contrast, temporal learning algorithms, such as ours, only need to address scenarios where a specific neuron fires a spike and records pertinent information. During the forward stage, a spike fired by an input neuron influences the state of both itself and all output neurons, culminating in a total of $O(C_{out})$ operations. During the backward stage, a spike discharged by an output neuron requires propagating gradient information to all intervening spikes between this spike and the last spike fired by the same neuron (thus, all input spikes are processed once during this stage). Therefore, by denoting the average firing rate of input and output neurons as $\alpha$ and $\beta$, the number of operations for this layer is determined as $O(T(\alpha C_{in}C_{out}+\beta C_{in}))$.
>
> In the case of sparse spikes, event-based learning algorithms exhibit an advantage, as $\alpha+\beta \ll 1$ under such circumstances.
>
>
> > Authors do not list BPTT-based methods in experiments. They say BPTT-based methods incorporate information when neurons do not fire in training, which improves the final performance. Does it mean BPTT-based methods have a better performance than time-based methods?
>
> We would like to point out that the performance of time-based training methods has yet to match that of surrogate gradient methods, attributed to the following reasons:
>
> Firstly, our event-driven approach restricts information propagation solely through spikes during both forward and backward propagation, as depicted in Figure 1d. In contrast, surrogate gradient techniques propagate gradient information even when no spikes are emitted, as exemplified in Figure 1b, where the surrogate gradient approximates $\frac{\partial s}{\partial u}$ regardless of spike occurrence. This event-driven property renders our method more challenging to train in comparison to surrogate gradient techniques, primarily due to the sparse gradient propagation path.
>
> Furthermore, our learning scheme is relatively novel in contrast to surrogate gradient methods.
>
> Conversely, the time-based training method boasts certain advantages over surrogate gradient methods. Specifically, our event-driven learning scheme exhibits enhanced biological plausibility, proving more amenable to continuous-time learning, and significantly more efficient when executed on neuromorphic hardware. The in-depth rationale is elucidated in the preceding question.

---

### Decision · Program_Chairs · 2023-09-21

**Decision:**

Accept (spotlight)

**Comment:**

The authors first analyze rate-based and time-based loss functions for the training of spiking neural networks (SNNs). Based on this, they propose an enhanced counting loss of SNNs as well as a new normalization method that tunes thresholds. They demonstrate in experiments that this approach outperforms previous time-based methods.

The reviewers acknowledge that the manuscript explores an interesting topic in SNN research. The manuscript provides detailed theoretical derivations, it is technically solid. The reviewers were convinced by the experiments, showing that the method outperforms previous time-based training methods.

A few questions regarding clarifications, the relation to BPTT-based methods, and the suitability for neuromorphic implementations have been addressed adequately in the rebuttal.

In summary, the manuscript presents a very nice study that improves the state-of-the-art for timing-based training of SNNs.